# Knowledge and attitudes of Lebanese women towards Baby Friendly Hospital Initiative practices

Hala Oueidat, Lama Charafeddine, Hana Nimer, Hiba Hussein, Mona Nabulsi[ID]*

Department of Pediatrics and Adolescent Medicine, American University of Beirut, Beirut, Lebanon

* mn04@aub.edu.lb

## Abstract

### Background

The World Health Organization and United Nations Children's Fund launched the Baby Friendly Hospital Initiative (BFHI) to encourage best infant breastfeeding practices immediately after birth. In Lebanon, few hospitals are currently accredited as Baby Friendly.

### Aim

To assess the knowledge of Lebanese women of BFHI steps, and to explore their attitudes towards Baby Friendly Hospitals, Skin-to-Skin Contact and Kangaroo Care practices.

### Methods

A cross-sectional survey of a random sample of healthy pregnant women from Lebanon's six governorates.

### Results

The mean (SD) age of the participants (N = 517) was 28.6 (4.7) years. Most participants were unfamiliar with the terms Baby Friendly hospital (93.7%), skin-to-skin contact or kangaroo care (75%), or were inadequately instructed on how to initiate (54.2%) or continue (46.2%) breastfeeding. However, when provided with information about the benefits of BFHI practices, most mothers (> 90%) stated that they would deliver in Baby Friendly hospitals. About 68.4% of mothers refused to give donor human milk to their sick premature infants because of religious beliefs. Knowledge of Baby Friendly hospitals was significantly associated with university education ($p = 0.029$), higher monthly income ($p = 0.042$), and previous experiences of skin-to-skin contact ($p<0.001$), rooming in ($p = 0.037$), or breastfeeding support ($p = 0.036$).

### Conclusion

There is a need for national awareness campaigns that address both the numerous advantages of the BFHI practices and Lebanese women's knowledge gaps about these practices. Such knowledge will help scale up the implementation of BFHI practices in hospitals in

**Data Availability Statement:** All relevant data are within the supporting information files.

**Funding:** The authors received no specific funding for this work.

**Competing interests:** The authors have declared that no competing interests exist.

Lebanon, thus increasing breastfeeding rates and positively impacting the health of infants and mothers.

## Introduction

Breastfeeding provides the newborn with the ideal nutrition that secures optimal growth and development. Exclusive breastfeeding is an effective public health measure that can reduce under-five mortality and morbidity in developing countries [1]. The World Health Organization (WHO) and the United Nations Children's Fund (UNICEF) global public health recommendation is to exclusively breastfeed infants for the first six months, and to continue breastfeeding for at least two years, supplemented with complementary foods that are nutritionally safe and adequate [2]. The WHO and UNICEF published in 1989 their "Ten Steps to Successful Breastfeeding" for the support of breastfeeding initiation and continuation [3]. This was followed in 1990 by the Innocenti Declaration which was developed by policy makers from forty countries, to encourage the protection, promotion and maintenance of breastfeeding [4]. Subsequently in 1991, the Baby Friendly Hospital Initiative (BFHI) with its ten steps was launched. To be accredited as 'Baby Friendly' by WHO/UNICEF, a maternity facility has to report at least 75% exclusive breastfeeding rate among mothers at discharge, to adhere to the International Code of Marketing Breast-milk Substitutes, and to successfully implement the "Ten Steps to Successful Breastfeeding". These steps are: "1) Have a written breastfeeding policy that is routinely communicated to all health care staff, 2) Train all health care staff in skills necessary to implement this policy, 3) Inform all pregnant women about the benefits and management of breastfeeding, 4) Help mothers initiate breastfeeding within a half-hour of birth, 5) Show mothers how to breastfeed, and how to maintain lactation even if they should be separated from their infants, 6) Give newborn infants no food or drink other than breast milk unless medically indicated, 7) Practice rooming in—allow mothers and infants to remain together—24 hours a day, 8) Encourage breastfeeding on demand, 9) Give no artificial teats or pacifiers (also called dummies or soothers) to breastfeeding infants, 10) Foster the establishment of breastfeeding support groups and refer mothers to them on discharge from the hospital or clinic" [3, 4]. Recently, the WHO reported an analysis of the current BFHI status in 168 countries around the world after 25 years of BFHI initiation [5]. As of 2016, the estimated overall BFHI coverage is 10%, with the highest rate being in the European region (36%), and the lowest in Africa and Southeast Asia (< 5%). The Eastern Mediterranean region (EMR) has an intermediate coverage of 17% that is largely driven by a few countries with large populations. Only one in five countries had ever accredited more than half of their facilities as Baby Friendly. For Lebanon, this rate declined from 18% to 10% in the past 5 years [5]. There is a great variation in BFHI implementation in the EMR countries based on the countries' categorization with regards to nutrition stages. Countries in the advanced nutrition stage like the Gulf Cooperation Council have 80% of their hospitals accredited as Baby Friendly, whereas countries in early nutrition transition like Lebanon have only 6.85% of their hospitals as Baby Friendly [6].

There is strong evidence to support the effectiveness of BFHI practices in improving breastfeeding rates. A Cochrane review of early skin-to-skin contact between mothers and their healthy neonates reported a 27% increase in breastfeeding between one and four months, as well as increased duration of breastfeeding [7]. Similarly, another Cochrane review of kangaroo mother care in preterm infants reported significant reduction in the risks of poor neonatal

outcomes such as mortality, sepsis, or hypothermia, as well as improved infant growth, breast-feeding, and maternal-infant bonding [8]. Implementation of nine of the ten BFHI steps in one cluster-randomized clinical trial from Congo significantly increased the rate of exclusive breastfeeding at six months, as compared to control [9]. Countries that increased the number of their Baby Friendly hospitals like Switzerland succeeded in increasing breastfeeding rates and duration. Women who delivered in Baby Friendly hospitals that were fully compliant with recommended UNICEF breastfeeding practices had longer duration of breastfeeding as compared to those delivering in less compliant Baby Friendly hospitals. Interestingly, breastfeeding rates also improved in non- Baby Friendly hospitals due to the public training programs and the wide availability of lactation counselors [10].

Lebanon is a country that suffers from low breastfeeding rates. Whereas initiation rate is high at 96% [11], exclusive breastfeeding in infants below six months drops to 15% [12]. Previous research from Lebanon identified several barriers to breastfeeding. These include cultural misconceptions such as breastfeeding causing weight gain, insufficiency of breast milk, maternal exhaustion or sleep deprivation, maternal employment, lack of familial support, lack of credible professional advice [13, 14], lack of recommended WHO and UNICEF training of health professionals in the prevention and treatment of breastfeeding problems [15, 16], as well as poor dissemination, implementation, and enforcement of policies and laws that protect breastfeeding [17].

In order to protect, promote and support breastfeeding in the community, the Lebanese Ministry of Health and Social Services, in collaboration with UNICEF and WHO, launched the BFHI in Lebanon in 1991. Despite governmental efforts to implement the ten BFHI steps, very few hospitals in Lebanon have been accredited as "Baby Friendly". The barriers to the implementation of the BFHI ten Steps are multifaceted. For example, obstetricians and pediatricians alike tend to easily advise mothers to supplement, or even substitute breastfeeding with artificial milk whenever mothers face any difficulty with breastfeeding. Health care workers in maternity facilities are reluctant to apply the skin-to-skin care immediately after delivery for fear of exposing the baby to cold environments or to accidental fall, as adequate supervision is not always possible. Rooming-in is a challenge because many women prefer to rest after delivery and delegate the care of their newborn to the grand-mother or hospital staff (personal observation), and referral to professional lactation experts or breastfeeding support groups is rarely done. Moreover, women may have several misconceptions regarding breastfeeding initiation and continuation [13, 14]. We have observed that most Lebanese women are unfamiliar with the term *Baby Friendly Hospital*, and are unaware of its practices, especially the *Skin-to-Skin Contact* or *Kangaroo Care* practices. Lebanese women tend to conform to social norms and standards dictated by their family and friends. Breastfeeding is therefore influenced by these norms as a new mother considers herself a member of this group, and her behavior is guided by her interactions with them. These subjective norms constitute the expectations of significant others with regards to one's act and the motivation to conform to these expectations [18].

This study aims to assess the knowledge of Lebanese pregnant women of BFHI steps, and to explore their attitudes towards *Baby Friendly Hospitals*, and acceptance of *Skin-to-Skin Contact* and *Kangaroo Care practices*. Information derived from this study will help tailor interventions to improve the awareness of Lebanese women about BFHI, and its advantages to the health of infants and mothers. It will also help scale up the implementation of BFHI practices in hospitals in Lebanon.

## Materials and methods

### Design

This study is a cross-sectional survey of Lebanese pregnant women about their knowledge and perceptions of BFHI. The study protocol was approved by the Institutional Review Board (IRB) of the American University of Beirut. All participants provided verbal informed consent prior to participation in the survey. The IRB waived the requirement for a written informed consent since the survey was anonymous and did not include any identifying data.

### Setting

Obstetric Clinics were chosen randomly from the list of the Lebanese Society of Obstetrics and Gynecology. The list includes the addresses and contacts of the 258 obstetric clinics from all six governorates (Mouhafaza) of Lebanon: Greater Beirut (Lebanon's capital), North Lebanon, South Lebanon, Mount Lebanon, Nabatiyeh, and Beqaa. A computer-generated random sampling of 20% of these clinics was conducted by an independent biostatistician. Thus 25 clinics from Greater Beirut and 5 clinics from each of the remaining 5 governorates were included in this study. This distribution was chosen so as to parallel the distribution of clinics in the different governorates of Lebanon.

### Sample

Recruitment of participants started in April 2016 and ended in March 2018. A trained research assistant contacted the randomly chosen clinics in each Mouhafaza to arrange for appointments to visit clinics for recruitment of participants. From each clinic, ten healthy pregnant women who were 18 years of age and older were consecutively recruited as they presented to the clinic. Recruitment of participants was happening in the same time frame for clinics located in the same Mouhafaza. The total sample size for this survey was 500 participants.

### Data collection

After getting their verbal consent, the participants self-administered an anonymous questionnaire to collect information about sociodemographic variables such as age, education, occupation, household monthly income, governorate name, gestational age, number of children, and number of breastfed children. The questionnaire asked participants about their knowledge of Baby Friendly Hospitals, and whether they had previous experiences with BFHI practices such as breastfeeding prenatal education, breastfeeding within the first hour, and rooming-in. The questionnaire included open ended questions that explored participants' attitudes towards BFHI practices such as rooming-in, exclusive breastfeeding from the first hour, pacifier use, skin-to-skin contact, previous experiences with skin-to-skin contact, breastfeeding a sick preterm newborn, or feeding her pumped mother's milk. Once participants had responded to the aforementioned questions, the trained research assistant briefed them about early breastfeeding and skin-to-skin contact benefiting the sick preterm infant health and shortening hospital stay. The participants then continued answering the remaining questions in the questionnaire that asked whether they would do skin-to-skin contact in the future, give pumped mother's own milk to the newborn, or accept to give her donor human milk. The final questions explored the participants' attitudes towards communication with certified lactation specialists, or referral to breastfeeding support groups (S1 and S2 Appendices).

## Data analysis

Univariate analysis of participants' baseline characteristics was conducted by summarizing continuous variables as means and standard deviations, or medians and interquartile ranges as appropriate, and summarizing categorical variables as counts and proportions. Bivariate analysis was conducted to examine the association between knowledge of Baby Friendly hospitals (as a surrogate for knowledge of BFHI practices) and each of the sociodemographic variables, previous breastfeeding experiences, and BFHI practices, using Student's *t* test for continuous variables and Pearson Chi-square test for categorical variables. Qualitative data that were generated from open-ended questions were subjected to content analysis and summarized by grouping them under the following themes: what a Baby Friendly hospital is, sources of information/ support of breastfeeding, reasons for choosing (or not) to deliver in hospitals that only allow breastfeeding, reasons for choosing (or not) to deliver in hospitals that ban pacifier use, what skin-to-skin contact is, descriptions of a previous skin-to-skin contact experience, reasons for choosing (or not) to repeat/perform skin-to-skin contact in the future, reasons for choosing (or not) to deliver in a hospital that implements skin-to-skin contact, reasons for continuing (or not) to breastfeed a hospitalized infant, reasons for willingness (or not) to pump mother's own milk to feed her premature newborn, reasons for willingness (or not) to put a premature newborn on the breast to feed her, reasons for willingness (or not) to breastfeed (or pump breast milk) a newborn within one hour from birth, reasons for accepting (or not) to give donor's milk to the newborn, reasons for willingness (or not) to communicate with a lactation consultant/ breastfeeding support group in future deliveries. The IBM-Statistical Package for Social Sciences (SPSS) version 23 was used for data entry, management, and analysis. Statistical significance was set at *p*-value < 0.05.

## Results

### Baseline characteristics

We enrolled 517 participants from the six governorates of Lebanon. About half of the participants had more than one child, were house makers, with a monthly household income below 1000 USD, and with university education. Table 1 details the baseline characteristics of the study's cohort.

The remaining quantitative results are reported together with the content analysis of the qualitative data so as to provide a narrative of the key findings.

### Knowledge about BFHI practices

The majority of participants were unfamiliar with the BFHI concept, including the practice of skin-to-skin contact. Those who reported previous knowledge of BFHI stated that Baby Friendly hospitals "*encourage breastfeeding and do not offer artificial milk to newborns*", "*keep mothers and their infants together*", and "*practice skin-to-skin contact*". Most women had knowledge of breastfeeding benefits but only about half of them were ever counseled about common breastfeeding problems. Similarly, half of the participants were informed about how to maintain breastfeeding at home or after resuming employment (Table 2). Counseling for information about breastfeeding was mostly done by their health care providers, such as doctors, nurses, midwives, International Board Certified Lactation Consultants, nutritionists, or by family members. A minority of participants reported getting their information from reading books, or from social media and the internet.

**Table 1. Baseline characteristics (N = 517).**

| Characteristic | Value |
|---|---|
| **Age (years)** | |
| Mean (SD) | 28.6 (4.7) |
| **Gestation (months)** | |
| Mean (SD) | 5.5 (2.5) |
| *__Number of children__ | |
| Median (IQR) | 1.0 (0.0–1.25) |
| *__Number of breastfed children__ | |
| Median (IQR) | 0.0 (0.0–1.0) |
| **Governorate, n (%)** | |
| Greater Beirut | 265 (51.3) |
| North Lebanon | 51 (9.9) |
| South Lebanon | 49 (9.5) |
| Nabatiyeh | 54 (10.4) |
| Mount Lebanon | 50 (9.7) |
| Bekaa | 48 (9.3) |
| **Employment, n (%)** | |
| Yes | 230 (49) |
| NO | 285 (55.3) |
| **Highest education, n (%)** | |
| ≤ Elementary | 17 (3.4) |
| Middle | 75 (15.1) |
| Secondary | 91 (18.3) |
| University | 313 (63.1) |
| **Household monthly income ($)** | |
| < 500 | 41 (11.2) |
| 500–1000 | 200 (54.5) |
| 1001–5000 | 114 (31.1) |
| > 5000 | 12 (3.3) |

* For multiparous participants only (N = 286).

## Experiences with hospital BFHI practices

The majority of multiparous participants had breastfed after giving birth to their last child in a hospital. However, only half of them did so within the first hour postpartum, or were instructed by a hospital health care worker on how to initiate and maintain breastfeeding. Maternal breast milk was the first feed given to infants in only 61.7% of participants. The majority of participants did not experience skin-to-skin contact within the first hour of giving birth. Moreover, they were separated from their infants after birth, and did not practice rooming in (Table 2). Participants who reported knowledge of skin-to-skin contact defined it as the "*immediate and direct contact between the mother and her infant*". They stated that it is beneficial as it promotes bonding, affection, and attachment to the baby, and makes the baby feel safe and secure. Few of these mothers reported other benefits like promoting breastfeeding, bolstering the immunity of the baby especially if sick, or that it helps regulate the infant's temperature. Only one mother reported that it relieves delivery pain. The few participants who reported a previous experience with skin-to-skin contact or rooming-in stated that it was "*a beautiful amazing experience*", and that they would be eager to repeat it. They felt affection,

**Table 2. Survey questions and participants responses (N = 517).**

| Question | Answer | |
|---|---|---|
| | n (%) | |
| | Yes | No |
| 1. Do you know what Baby Friendly Hospitals are? | 32 (6.3) | (93.7) |
| 2.*In your previous pregnancies, did anyone talk to you about the benefits of breastfeeding? | 221 (80.1) | 55 (19.9) |
| 3.*In your previous pregnancies, did anyone counsel you about common problems during breastfeeding or about management of breastfeeding when going back to work or home? | 149 (53.8) | 128 (46.2) |
| 4.*When you delivered your last baby, did you breast feed after birth? | 201 (72.6) | 76 (27.4) |
| 5.*When you delivered your last baby, did anyone help you start breastfeeding in the first hour? | 124 (45.8) | 147 (54.2) |
| 6.*When you delivered your last baby, did anyone show you how to breastfeed your baby? | 128 (47.2) | 143 (52.8) |
| 7.*Did anyone inform you how to maintain breastfeeding when you are away from your baby? | 131 (50.4) | 129 (49.6) |
| 8.*What was the first food or drink that your previous baby had soon after delivery? | | |
| ▪ Breast milk | 166 (61.7) | N/A |
| ▪ Formula | 96 (35.7) | |
| ▪ Dextrose | 7 (2.6) | |
| 9.*When you delivered you previous baby, did the baby stay with you in the same room all the time (24 hours)? | 51 (19.0) | 218 (81.0) |
| 10. Would you like to deliver in a hospital in which the baby and the mother are kept in the same room? | 428 (85.1) | 75 (14.9) |
| 11. Would you deliver your next baby in a hospital that encourages *breastfeeding on demand*, and gives formula only if medically indicated? | 461 (90.7) | 47 (9.3) |
| 12. Would you deliver in a hospital that bans pacifiers? | 348 (68.6) | 159 (31.4) |
| 13. Do you know about *Skin-to-Skin Contact* between mother and baby? | 128 (25.0) | 383 (75.0) |
| 14. Did you have any previous experience with skin-to-skin contact? | 69 (14.0) | 424 (86.0) |
| 15. Would you be willing to deliver in a hospital that practices skin-to-skin contact? | 404 (91.4) | 38 (8.6) |
| 16. If the baby was ill and admitted to the hospital for any reason, would you be willing to breastfeed her? | 490 (96.3) | 19 (3.7) |
| 17. If you delivered prematurely, would you be willing to pump breastmilk to give to your baby? | 479 (95.8) | 21 (4.2) |
| 18. If you delivered prematurely, would you be willing to put the baby to breast if the doctor allows it? | 487 (97.8) | 11 (2.2) |
| 19. Would you do skin-to-skin contact with your premature baby? | 473 (97.7) | 11 (2.3) |
| 20. Would you initiate breast feeding or administer pumped milk to your premature baby early? | 469 (95.3) | 23 (4.7) |
| 21. Would you be willing to pump breast milk within the first hour after delivery? | 316 (68.8) | 143 (31.2) |
| 22. Would you accept breast milk from a donor to be given to your baby? | 158 (31.6) | 342 (68.4) |
| 23. Would you like to communicate with a lactation specialist as soon as you deliver? | 386 (79.4) | 100 (20.6) |
| 24. Would you be willing to be referred to a breastfeeding support group after being discharged? | 401 (82.2) | 87 (17.8) |

*Questions 2 to 9 were answered by multiparous women only (N = 286).

tenderness and love towards their babies. They also felt their infants to be more secure, safe and easily soothed. Moreover, it relieved their pain and elevated their feelings of bonding and motherhood, and promoted breastfeeding. One mother described it as "..*a gift from God*..".

### Attitudes towards BFHI practices

With respect to attitudes relating to BFHI practices, most participants stated that they would like to deliver their next infant in a hospital that practices skin-to-skin contact, whether the infant was full-term or premature. They would do skin-to-skin contact as much as advised or allowed by their doctors. Moreover, they would prefer that their infants be roomed-in with them during hospital stay (Table 2). Participants who rejected the idea reported that they lacked the knowledge about these practices, would not do it out of fear that their health, or that of the infant is compromised by skin-to-skin contact, or because Lebanon lacks hospitals that practice rooming-in and skin-to-skin contact. As for delivering in a hospital that bans pacifier use for newborns, 68.6% of mothers indicated that they would deliver in such hospitals. Their reasons were that use of pacifiers may promote infections in the infant, affect breastfeeding negatively, and may harm babies' teeth development. Participants who preferred not to deliver in hospitals that ban pacifier use stated that the pacifier is beneficial to the baby as it soothes her, and relieves the mother.

Hospitals that encourage breastfeeding on demand, and give artificial milk only if medically indicated were favored by the majority of mothers. They stated that breastfeeding has many benefits such as improving the babies' immunity and protecting her from infections, providing better nutrition and vitamins to the infant, and strengthening the bond with the infant. In special situations like having a sick hospitalized infant, or a premature newborn, most mothers preferred to continue breastfeeding their infants during hospitalization, initiate early breastfeeding for the premature newborn, or provide her with pumped mother's own milk. However, only 68.8% stated that they would initiate pumping in the first hour postpartum, and preferred direct breastfeeding believing that it is more nutritious, and provides better immunity to their babies. Only a third approved of giving donor human milk to their sick premature infants, with the majority opposing the idea because of religious beliefs, fear of a lower quality of the donor milk, fear of transmitting infections to the baby from the donor mother, or fear that their infants may become attached to the donor mother. Instead, they preferred resorting to artificial milk as alternative to donor human milk (Table 2).

As for utilizing available resources for breastfeeding support, the majority of participants stated that they would be interested in communicating with a lactation consultant soon after giving birth, and/or be referred to a breastfeeding support group.

In bivariate analysis, maternal knowledge of Baby Friendly hospitals was significantly associated with university education, higher household monthly income, having previous positive experiences with skin-to-skin contact or rooming-in, and getting instructions about breastfeeding initiation and maintenance (Table 3).

## Discussion

This cross-sectional study is the first one from Lebanon to explore maternal knowledge of, and attitudes towards BFHI practices among Lebanese mothers. Our cohort was randomly chosen from all six governorates of Lebanon; hence the findings of this study are representative of Lebanese women in the reproductive age group. In view of the low breastfeeding rates reported from Lebanon, this study is timely as it sheds light on some societal barriers to BFHI implementation.

**Table 3. Bivariate analysis (N = 511).**

| Continuous Variables | Knowledge of Baby Friendly Hospitals | | |
|---|---|---|---|
| | Yes (n = 32) | No (n = 479) | p-value |
| | M (SD) | M (SD) | |
| Age (years) | 29.6 (5.4) | 28.6 (4.7) | 0.254 |
| Gestational Age (months) | 4.9 (2.6) | 5.5 (2.5) | 0.229 |
| Number of children* | 1.6 (0.6) | 1.8 (1.0) | 0.283 |
| Number of breastfed children* | 1.5 (0.7) | 1.5 (1.1) | 0.775 |
| **Categorical Variables** | **Yes (n = 32)** | **No (n = 479)** | **p-value** |
| | n (%) | n (%) | |
| University education | 26 (81.3) | 284 (62.0) | 0.029 |
| Monthly household income >1000 USD | 15 (51.7) | 111 (33.0) | 0.042 |
| *Previously informed about breastfeeding benefits | 17 (89.5) | 201 (79.1) | 0.278 |
| *Previously instructed about how to maintain breastfeeding | 14 (77.8) | 134 (52.3) | 0.036 |
| *Had breastfed immediately after delivery | 17 (89.5) | 181 (71.0) | 0.082 |
| *Previously supported to initiate breastfeeding during the first hour | 12 (63.2) | 111 (44.6) | 0.117 |
| *Received previous training in breastfeeding | 11 (57.9) | 116 (46.6) | 0.341 |
| *Received previous training on how to sustain breastfeeding | 14 (73.7) | 116 (48.7) | 0.036 |
| *Previous child fed maternal milk as the first feed | 13 (72.2) | 150 (60.5) | 0.324 |
| *Practiced rooming-in in previous delivery | 7 (36.8) | 43 (17.4) | 0.037 |
| Will deliver in a hospital that practices rooming-in (Yes) | 27 (84.4) | 395 (84.9) | 0.930 |
| Will deliver in a hospital that practices EBF | 26 (83.9) | 429 (91.1) | 0.182 |
| Will deliver in a hospital that bans pacifier use | 17 (53.1) | 327 (69.7) | 0.050 |
| Has knowledge of skin-to-skin contact | 22 (68.8) | 103 (21.7) | <0.001 |
| Has previous experience of skin-to-skin contact | 13 (41.9) | 56 (12.3) | <0.001 |
| Will deliver in a hospital that practices skin-to-skin contact (Yes) | 27 (96.4) | 374 (91.0) | 0.323 |
| Will continue breastfeeding hospitalized newborn | 30 (93.8) | 454 (96.4) | 0.448 |
| Will give pumped maternal milk to premature newborn | 31 (96.9) | 442 (95.7) | 0.744 |
| Will breastfeed premature newborn immediately after birth | 30 (93.8) | 451 (98.0) | 0.112 |
| Will do skin-to-skin contact with premature newborn | 31 (96.9) | 437 (97.8) | 0.746 |
| Will breastfeed early on or pump milk | 30 (96.8) | 434 (95.4) | 0.719 |
| Will pump breasts in the first hour postpartum | 19 (61.3) | 294 (69.5) | 0.340 |
| Accepts to give donated human milk to newborn | 9 (28.1) | 146 (31.6) | 0.682 |
| Future communication with a lactation consultant (Yes) | 22 (71.0) | 362 (80.1) | 0.224 |
| Future communication with breastfeeding support groups (Yes) | 28 (90.3) | 369 (81.5) | 0.214 |

BFHI = Baby Friendly Hospital Initiative; M = Mean; SD = Standard deviation; EBF = Exclusive breastfeeding

*In multiparous participants, N = 286

Our study reveals that Lebanese mothers have poor knowledge of BFHI, its recommended practices, and their benefits to the infant and mother, which represents an important barrier to BFHI implementation. Participants with knowledge of Baby Friendly hospitals and a previous experience of skin-to-skin contact, or rooming-in, or those who were well-instructed about breastfeeding were very willing to repeat the same experience in the future. Interestingly, most mothers with poor knowledge of BFHI practices stated that they would deliver their future infants in Baby Friendly hospitals after they were provided with the information about the benefits of these practices.

The majority of our participants would continue breastfeeding their infants if hospitalized, and would initiate early breastfeeding or give pumped maternal milk to their sick premature

newborns. However, mothers who stated otherwise justified their choices by fear that continuing breastfeeding would compromise the infant's health status. Some participants rejected the idea of feeding pumped maternal milk to the infant believing that breastfeeding is better than pumped maternal milk. Interestingly, mothers would consider breastfeeding a sick infant, or express milk early on to give to a sick premature if so advised by their doctors. This highlights the important role of physicians in advocating for early initiation of breastfeeding in such infants, as well as continuing to breastfeed sick hospitalized infants. A previous qualitative study from Lebanon found that a pregnant woman has full trust in her obstetrician as she puts her life in his/her hands. The doctor's advice, even if conflicting with her own beliefs will make her question her own beliefs and act in accordance with her doctor's advice [14]. A recent study that explored whether breastfeeding initiation and duration would differ by prenatal care provider reported that women who received counseling from a midwife were more likely to exclusively breastfeed as compared to those who received prenatal care from an obstetrician [19]. Hence, there is a need to involve obstetricians in the efforts to promote breastfeeding very early during pregnancy, as part of routine prenatal care. In fact, the American College of Obstetricians and Gynecologists' current recommendation is for obstetricians to advocate for, and support breastfeeding when offering prenatal care [20]. In addition, all health care workers should be educated about BFHI practices and their benefits to maternal and infant health. It has been previously shown that providing breastfeeding education to hospital nursing staff would positively affect their breastfeeding beliefs, their compliance with BFHI, and would increase in-hospital exclusive breastfeeding rate [21].

Our study reveals interesting cultural beliefs about the use of donor human milk to feed infants when maternal milk is unavailable. The first belief concerns acceptance of donor milk: although all participants were knowledgeable of the benefits of human milk, only one third would accept to give donor milk. This was based on a religious misconception that feeding donor human milk to the infant is banned in the Muslim religion. In fact, neither breastfeeding another woman's infant, nor feeding the infant donor human milk are banned in the Muslim religion. In Islam, an infant who is fed human milk from a woman other than his biologic mother cannot in the future marry from the offspring of the donor mother, as the donor's children would be considered his/her siblings. If the donor is unknown to the biologic mother, then the parents of the recipient child would reject donor milk out of fear that their infant may later marry a sibling, even though it is a remote possibility. Religious beliefs are considered strong social barriers to human milk donation in Islamic countries, where sharing a woman's milk creates kinship and constitutes an impediment to marriage [22–24]. However, several Islamic countries reported success stories with regards to wet nursing by establishing milk banks that respect the Islamic rules of donor milk and marriage [25, 26]. The other cultural belief pertaining to human donor milk is fear that it may have a 'lower quality', or that it may be an infectious hazard to the infant.

It is interesting to note that many mothers referred to their families as a source of breastfeeding information, similar to what has been reported from other countries in the region [27, 28]. Cues that a mother receives from her family and friends become her main source of information when back home after delivery. Thus familial support is essential for mothers as the social norms dictated by the woman's family may influence her breastfeeding habits [14, 27, 28]. Since the majority of participants would like to communicate with a lactation consultant soon after giving birth, having lactation consultants in hospitals for breastfeeding support is crucial to the establishment and continuation of breastfeeding. A recent trial by our group revealed that mothers who received a multi-component breastfeeding intervention consisting of breastfeeding education by a lactation expert, peer, and professional lactation support were six times more likely to practice exclusive breastfeeding for six months, as compared to

standard care [29]. Whereas peer support was viewed by the trial's participants as important in encouraging them to continue breastfeeding, lactation expert support was considered influential in helping them solve technical problems with breastfeeding [30]. Although the presence of certified lactation consultants is recommended by the Ministry of Public Health, this recommendation is not a requirement by the government for hospital accreditation.

Our study is limited by the fact that it did not explore in depth the identified barriers to BFHI implementation, which needs a qualitative methodology such as interviews or focus group discussions. However, the study may be viewed as a first step to address societal barriers to the implementation of BFHI in hospitals and maternities in Lebanon. Its main strength is its representation of the Lebanese population, since the sampling frame was a simple random sample from all governorates of Lebanon.

In conclusion, this study highlights the need for national awareness campaigns that address both the numerous advantages of the BFHI practices and Lebanese women's knowledge gaps about these practices. Such knowledge will facilitate the implementation of BFHI in Lebanon as women may become more engaged in these practices, and will seek to deliver in Baby Friendly hospitals thus encouraging maternity facilities to adopt the BFHI practices. Moreover, there is a need to spread the knowledge about BFHI practices to the public at large since family and peers may be influential in breastfeeding practices, as well as to obstetricians, pediatricians and family physicians that are the primary source of medical information for patients. In-depth qualitative research is needed to further explore the identified barriers.

## Supporting information

**S1 Appendix. English questionnaire.**
(PDF)

**S2 Appendix. Arabic questionnaire.**
(PDF)

**S1 Dataset. Anonymized data set.**
(SAV)

## Author Contributions

**Conceptualization:** Lama Charafeddine, Hiba Hussein, Mona Nabulsi.

**Data curation:** Hala Oueidat, Hana Nimer, Hiba Hussein, Mona Nabulsi.

**Formal analysis:** Hala Oueidat, Mona Nabulsi.

**Investigation:** Hala Oueidat, Hana Nimer, Mona Nabulsi.

**Methodology:** Hiba Hussein, Mona Nabulsi.

**Project administration:** Mona Nabulsi.

**Resources:** Mona Nabulsi.

**Software:** Mona Nabulsi.

**Supervision:** Mona Nabulsi.

**Validation:** Mona Nabulsi.

**Visualization:** Lama Charafeddine, Mona Nabulsi.

**Writing – original draft:** Hala Oueidat, Lama Charafeddine, Hana Nimer, Mona Nabulsi.

**Writing – review & editing:** Hala Oueidat, Lama Charafeddine, Hana Nimer, Hiba Hussein, Mona Nabulsi.

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
