## [Decision Letter · Decision Letter 0]

19 Jun 2020

PONE-D-20-05601

Knowledge and attitudes of Lebanese women towards Baby Friendly Hospital Initiative

PLOS ONE

Dear Dr. Nabulsi,

Thank you for submitting your manuscript to PLOS ONE. After careful consideration, we feel that it has merit but does not fully meet PLOS ONE’s publication criteria as it currently stands. Therefore, we invite you to submit a revised version of the manuscript that addresses the points raised during the review process.

We look forward to receiving your revised manuscript.

Kind regards,

Thach Duc Tran, M.Sc., Ph.D.

Academic Editor

PLOS ONE

Journal Requirements:

2. Please amend your current ethics statement to address the following concerns: Please explain why written consent was not obtained, how you recorded/documented participant consent, and if the ethics committees/IRBs approved this consent procedure.

3. Please include additional information regarding the survey or questionnaire used in the study and ensure that you have provided sufficient details that others could replicate the analyses. For instance, if you developed a questionnaire as part of this study and it is not under a copyright more restrictive than CC-BY, please include a copy, in both the original language and English, as Supporting Information.  If the original language is written in non-Latin characters, for example Amharic, Chinese, or Korean, please use a file format that ensures these characters are visible.

Reviewers' comments:

Reviewer's Responses to Questions

**Comments to the Author**

1. Is the manuscript technically sound, and do the data support the conclusions?

Reviewer #1: Yes

Reviewer #2: Yes

Reviewer #3: Partly

2. Has the statistical analysis been performed appropriately and rigorously? 

Reviewer #1: Yes

Reviewer #2: Yes

Reviewer #3: I Don't Know

3. Have the authors made all data underlying the findings in their manuscript fully available?

Reviewer #1: Yes

Reviewer #2: Yes

Reviewer #3: No

4. Is the manuscript presented in an intelligible fashion and written in standard English?

Reviewer #1: Yes

Reviewer #2: Yes

Reviewer #3: Yes

5. Review Comments to the Author

Reviewer #1: The article is well written with appropriate research methodology and the findings will be beneficial to the vitalization of BFHI in Lebanon.

One issue is the statement on lines 78 and 79 that the majority of BFHI hospitals are in Quebec, Canada. Baby-Friendly hospitals are distributed around the world with the highest number reported by country in China. Quebec may certainly have the most in Canada, but not the world!

Reviewer #2: Overall the paper has provided a good data with an appropriate analysis. As a second language, English language in this paper seems to be standard. I recommend some references from other Middle East countries should be added.

Reviewer #3: Responses to points above:

1. With further explanation of the method and analysis it is likely the paper will meet this criteria, but there are gaps at the moment. More specific comments are provided below.

2. The statistical test seems appropriate – however I have asked for clarification around the use of “knowledge of baby friendly hospitals” to be a marker for “knowledge of BFHI practices”. The aim of the study is stated refers to broader issues than knowing “what Baby Friendly hospitals are” so it is not clear why this is the only factor investigated in the bivariate analysis.

3. At this point all data is not available however the authors state that the data will be publicly available from PLOS ONE upon acceptance.

4. While I have ticked “yes” there are some minor grammatical errors that should be corrected by proof reading.

Thank you for the opportunity to review this paper on the Knowledge and attitudes of Lebanese women towards the Baby Friendly Hospital Initiative. This study presents the results of original research that as far as I am aware has not been published elsewhere.

The paper makes a compelling case for the research in the context of the current literature and in the face of low breastfeeding rates in Lebanon. It outlines the importance of breastfeeding and the established role of BFHI practices in increasing breastfeeding rates internationally. The methodology is appropriate for gaining knowledge of women’s views and beliefs and the findings support the need for national education aimed at improving knowledge of BFHI practices with a view to increasing breastfeeding within Lebanon. The study was approved by the Institutional Review Board of the American University of Beirut and all participants provided oral consent. This study builds on previous research in Lebanon on the subject.

The survey methodology is appropriate for answering the research question - including the mix of (apparently) yes/no responses and (apparently) the option to add comments. The detail of the survey requires further explanation as indicated below. An interesting aspect is that education was provided following phase one of the survey and then participants were asked to respond to further questions. The findings provide rich data on issues that could be addressed in national education to improve breastfeeding rates and thus the health of women and babies.

The paper is mostly well written and easy to follow, although there are some minor errors in grammar that require correction. The discussion captures the findings in relation to existing literature and explores issues raised in a clear and engaging way. In particular exploration of the cultural issues are very informative and open possibilities for ensuring messages are appropriately targeted. A strength of the study is that over 500 participants representing all six governorates (Mouhafaza) of Lebanon completed the questionnaire. The conclusion is based on the findings but needs rewording to ensure it aligns directly with the aims of the study – please see below.

I have identified broad areas that I suggest need expansion or restructuring to improve the paper.

1. Please consider providing further explanation about the BFHI practices, the 10 steps to successful breastfeeding, and the process of accreditation as a Baby Friendly hospital in the introduction. Two practices: skin to skin and kangaroo care have been targeted in the survey. It would be helpful to define these two practices and then to explain why they have been targeted in the survey as they only constitute one of the 10 steps. Was there a particular reason why you wanted to highlight these practices in your aim and not the other BFHI practices?

2. I suggest the conclusion, “Such knowledge will facilitate the implementation of BFHI in Lebanon” requires rephrasing. My understanding is that the hospitals would drive the process of implementing the BFHI practices but if women had a greater understanding they might more readily engage in these practices. This just needs a small modification of the language.

3. Methods: In general there is insufficient detail to understand the specifics of the consent process (why was this an oral process?), the precise nature of the question types, length of the questionnaire and administration of the questionnaires. Considerable detail is lacking from this section. Eg Please explain the nature of the open ended questions eg did they ask about benefits of breastfeeding etc

4. More detail about analyses is also required for both the statistical analysis and the content analysis – how was this completed?

5. Results: The results section could also be reported in more detail. The reader could better understand the findings if it was stated that they are reported with reference to both the quantitative data and the content analysis of the qualitative data to provide a narrative of the key findings. The purpose of Table two is clear but it would be helpful, as stated above, if a description of the questionnaire was included in the methods. This would then give context to the table. The bivariate analysis would benefit from more description. Line 239 refers to knowledge of BFHI practices while the question from the survey asks about knowledge of baby friendly hospitals. This must be consistent as I suggest the two things are not the same. Similarly Table 3 refers to BFHI practices while the question from the survey asks about knowledge of baby friendly hospitals. There is no rationale for why this question from the questionnaire was used for this particular analysis. The aim of the study is stated refers to broader issues than knowing “what Baby Friendly hospitals are” so it is not clear why this is the only factor investigated in the bivariate analysis. Further explanation is required.

6. Discussion is well written and situates the results in the context of the literature. There are very interesting cultural issues discussed which puts the findings in the local context. I suggest a further comment on the impact of the obstetrician’s views and the family views. Evidence is provided that shows both to be strongly influential. It would add value if the authors could comment on the possible conflict of differing views by medical staff and family and how a mother may resolve such conflict. Nabulsi et al (2019) describe the importance of combining antenatal breastfeeding education with peer and professional lactation support to improve breastfeeding rates. Perhaps this concept could be expanded in the current paper.

Specific points

7. Please state why written consent was not required.

8. It is not clear how the complete data set will be made available to meet the statement: The data will be publicly available from PLOS ONE upon acceptance.

9. Table 1 reports median (SD) for some items. I suggest median(IQR) or mean (SD).

10. Line 23 Introduction. I suggest changing the term designated as Baby Friendly to accredited as baby friendly – unless there is a reason for using the term designated.

11. Line 25 Aim. I suggest a rearrangement of the aim to improve understanding. The aim stated in the manuscript lines 116 – 117 is structured more appropriately. Please consider using this to replace the current aim in the abstract. Thank you

12. Abstract Results: minor grammatical errors

13. Line40- 41: the phrase “and their numerous advantages” does not make it clear what “their” is referring to and thus could be misleading. Suggest: “There is a need for national awareness campaigns that address both the numerous advantages of BFHI practices and Lebanese women’s knowledge gaps about these practices.”

14. Line 42 suggest adding and thus “through increased breastfeeding” positively impact

15. Please consider including the ideas expressed in lines 118 – 121 as part of the conclusion of the abstract.

16. Line 129 states clinics while line 130 states licensed obstetricians. Should clinics be the consistently used term here for consistency? As part of this description please state the process for random identification of the clinics and also the total number of clinics from which the 50 clinics were chosen. Thank you

17. Line 138; please state the criteria for consecutive recruitment eg when did this recruitment start and was it the same time frame for each clinic?

18. Line 141. Please clarify why verbal and not written consent was obtained. Also clarify if the questionnaire was written and if self administered did the participant take the questionnaire away or answer in the clinic. Was there any involvement of the staff? How long was the questionnaire expected to take and how many questions and what type of responses were included. Later in the section open ended questions are described. Generally more detailed information is required in this section. “Participants were asked” may be misleading could you rephrase to ”The questionnaire asked participants about…”

19. Line 150 were participants briefed after they completed the first part of the questionnaire? Please make explicit and include a statement about who briefed the participants eg Obstetrician etc

20. Line 152 again please clarify if these questions were verbally asked or if the statement refers to the written questions in the survey.

21. Line 157 I note medians are also included in Table 1but not stated in this section.

22. Line 255. Please clarify if it is only mothers with poor knowledge who would birth their baby in a Baby Friendly Hospital next time. I suspect this is about the phrasing of the sentence rather than the data as the next sentence describes this. Can you change the order of the sentences to make this clearer?

23. Line 313 Comment as for the conclusion in the abstract. The authors provide evidence that family and peers may be influential in breastfeeding practices and thus I suggest this is also addressed in terms of increasing their knowledge.

Many thanks.

6. PLOS authors have the option to publish the peer review history of their article (what does this mean?). If published, this will include your full peer review and any attached files.

Reviewer #1: No

Reviewer #2: No

Reviewer #3: No

---

## [Author Response · Author response to Decision Letter 0]

20 Jul 2020

July 20, 2020 

To: PLOS ONE Editor

Subject: “Knowledge and attitudes of Lebanese women towards Baby Friendly Hospital Initiative practices”

Dear Editor,

 We are submitting our revised manuscript PONE-D-20-05601 titled, “Knowledge and attitudes of Lebanese women towards Baby Friendly Hospital Initiative practices” for publication in PLOS ONE. All changes have been highlighted in yellow in the revised tracked copy. We are very thankful for the constructive comments of the reviewers which we hope will improve our manuscript. 

Below, please find our response to each comment.

Response to Editorial comments 

1. Ensure that the manuscript meets PLOS ONE’s style requirements, including those for file naming.

Answer: We reviewed and revised the manuscript to meet PLOS ONE’s style. We hope that you find no violations of the requirements. Should there be any that escaped our scrutiny, we would greatly appreciate if you can alert us of the specific deficiency in future communications so we address it immediately and correctly. 

2. Please amend your current ethics statement to address the following concerns: Please explain why written consent was not obtained, how you recorded/documented participant consent, and if the ethics committees/IRBs approved this procedure.

Answer: We requested from our IRB a waiver of written informed consent since the survey was anonymous and no identifying data was to be collected from the participants. The IRB approved the waiver of written informed consent. When participants were approached for enrolment in the study, they were informed about the purpose of the study and the type of information that would be collected. They were provided with a written form that summarized the study’s objective and procedure, as well as their rights as research participants and contacts of the research team and the IRB. Please find attached to this rebuttal a copy of the IRB’s approval letter and a copy of the oral consent form. We added the following paragraph to the Materials and Methods, Design section: “The study protocol was approved by the Institutional Review Board (IRB) of the American University of Beirut. All participants provided verbal informed consent prior to participation in the survey. The IRB waived the requirement for a written informed consent since the survey was anonymous and did not include any identifying data”. 

Answer: We submitted to PLOS ONE English and Arabic copies of the Questionnaire used in this study as Supporting Information (S1 and S2 Appendices). 

Answer: There are no ethical or legal restrictions on sharing our de-identified data set. We have uploaded our anonymized data set as Supporting Information File (S3 Dataset). Kindly update our Data Availability statement on our behalf to reflect this information.

Response to Reviewer #1

We greatly appreciate your positive comments and suggestions. Kindly find below our response. 

1. The statement on lines 78 and 79 that the majority of BFHI hospitals are in Quebec, Canada. Baby-Friendly hospitals are distributed around the world with the highest number reported by country in China. Quebec may certainly have the most in Canada, but not the world.

Answer: We removed this statement and added information about BFHI implementation in different areas of the world from the WHO 2017 report: “Recently, the WHO reported an analysis of the current BFHI status in 168 countries around the world after 25 years of BFHI initiation [5]. As of 2016, the estimated overall BFHI coverage is 10%, with the highest rate being in the European region (36%), and the lowest in Africa and Southeast Asia (< 5%). The Eastern Mediterranean region (EMR) has an intermediate coverage of 17% that is largely driven by a few countries with large populations. Only one in five countries had ever accredited more than half of their facilities as Baby Friendly. For Lebanon, this rate declined from 18% to 10% in the past 5 years [5]. There is a great variation in BFHI implementation in the EMR countries based on the countries’ categorization with regards to nutrition stages. Countries in the advanced nutrition stage like the Gulf Cooperation Council have 80% of their hospitals accredited as Baby Friendly, whereas countries in early nutrition transition like Lebanon have only 6.85% of their hospitals as Baby Friendly [6]”.

Response to Reviewer #2

Thank you for your positive feedback and valuable suggestion. Please find below our response. 

1. Some references from other Middle East countries should be added. 

Answer: We added a paragraph in the Introduction that summarizes the most recent (2017) WHO report about the BFHI status in different regions of the world, including the Eastern Mediterranean Region: “Recently, the WHO reported an analysis of the current BFHI status in 168 countries around the world after 25 years of BFHI initiation [5]. As of 2016, the estimated overall BFHI coverage is 10%, with the highest rate being in the European region (36%), and the lowest in Africa and Southeast Asia (< 5%). The Eastern Mediterranean region (EMR) has an intermediate coverage of 17% that is largely driven by a few countries with large populations. Only one in five countries had ever accredited more than half of their facilities as Baby Friendly. For Lebanon, this rate declined from 18% to 10% in the past 5 years [5]. There is a great variation in BFHI implementation in the EMR countries based on the countries’ categorization with regards to nutrition stages. Countries in the advanced nutrition stage like the Gulf Cooperation Council have 80% of their hospitals accredited as Baby Friendly, whereas countries in early nutrition transition like Lebanon have only 6.85% of their hospitals as Baby Friendly [6]”. 

Response to Reviewer #3

Thank you for your critical review of our manuscript and the valuable comments. Please find below our point by point response. 

1. With further explanation of the method and analysis it is likely that the paper will meet these criteria, but there are gaps at the moment. More specific comments are provided below.

Answer: Thank you. We have addressed the comments below and hope that our answers fill the gaps identified by the kind reviewer.

2. The statistical test seems to be appropriate – however I have asked for clarification around the use of “knowledge of Baby-Friendly hospitals” to be a marker for “knowledge of BFHI practices”. The aim of the study as stated refers to broader issues than knowing “what Baby-Friendly hospital are” so it is not clear why this is the only factor investigated in the bivariate analysis.

Answer: We agree with the kind reviewer that ‘Knowledge of Baby Friendly hospitals’ is not the same as ‘Knowledge of BFHI practices’. We thus changed the title of the manuscript to reflect the main aim of the study which is knowledge of, and attitudes towards BFHI practices. As for the bivariate analysis, we chose ‘knowledge of Baby Friendly hospitals’ as a surrogate for ‘knowledge of BFHI practices’ since we know from our daily practice with mothers that the great majority were unfamiliar with Baby Friendly hospitals and BFHI practices, and that those who knew what Baby Friendly Hospitals are, were also familiar with the BFHI practices and that these hospitals implement such practices. Except for the practice of breastfeeding, most mothers in Lebanon have never heard of skin-to-skin-contact or kangaroo mother care, and have never practiced rooming in. After careful review of the univariate analysis, we thought that ‘knowledge of Baby Friendly hospitals’ can serve as a good marker of ‘knowledge of BFHI practices’. We conducted several bivariate analyses between ‘skin-to-skin contact and other BFHI practices versus the remaining sociodemographic variables. We got very similar results to the bivariate analysis between ‘knowledge of Baby Friendly hospitals’ and the other variables. Hence we settled on ‘knowledge of Baby Friendly hospitals’ as the most appropriate surrogate marker of knowledge of BFHI practices for bivariate analysis.

3. At this point all data is not available however the authors state that the data will be publicly available from PLOS ONE upon acceptance.

Answer: We uploaded our anonymized data set as Supporting Information File (S3) on PLOS ONE submission system as part of this revision.

4. While I have ticked “yes” there are some minor grammatical errors that should be corrected by proof reading.

Answer: We proofread the revised manuscript and corrected the grammatical errors. 

5. Introduction: Please consider providing further explanation about the BFHI practices, the 10 steps to successful breastfeeding, and the process of accreditation as a Baby Friendly hospital in the introduction. Two practices: skin to skin and kangaroo care have been targeted in the survey. It would be helpful to define these 2 practices and then to explain why they have been targeted in the survey as they only constitute one of the 10 steps. Was there a particular reason why you wanted to highlight these practices in your aim and not the other BFHI practices? 

Answer: We added a paragraph in the Introduction that further explains the BFHI ten steps, and the process of Baby Friendly accreditation. We specifically targeted skin-to-skin contact or kangaroo care practices because most mothers in Lebanon have never heard of skin-to-skin-contact or kangaroo mother care, and have never practiced rooming in. Moreover, when we tried to convince some hospital administrators to implement these practices we were told that Lebanese women would not ‘like’ such practices as they may find them threatening to the well-being of their newborns. This was one of the major reasons why we conducted this survey as evidence to the contrary argument: that women would welcome such practices if they knew of their advantages. In the Introduction, we added the following sentence to clarify why we emphasized on skin-to-skin contact and kangaroo mother care: “We have observed that most Lebanese women are unfamiliar with the term Baby Friendly Hospital, and are unaware of its practices, especially the Skin-to-Skin Contact or Kangaroo Care practices.” 

6. I suggest the conclusion; “Such knowledge will facilitate the implementation of BFHI in Lebanon” requires rephrasing. My understanding is that the hospitals would drive the process of implementing the BFHI practices but if women had a greater understanding they might more readily engage in these practices. This just needs a small modification of the language.

Answer: Thank you. We added the following sentence to reflect this point: “Such knowledge will facilitate the implementation of BFHI in Lebanon as women may become more engaged in these practices, and will seek to deliver in Baby Friendly hospitals, thus encouraging maternity facilities to adopt the BFHI practices”.

7. Methods: In general there is insufficient detail to understand the specifics of the consent process (why was this an oral process?), the precise nature of the question types, length of the questionnaire and administration of the questionnaire. Considerable detail is lacking from this section. Eg. Please explain the nature of the open ended questions. Eg. Did they ask about benefits of breastfeeding etc..

Answer: With regards to the oral consent, we added the following paragraph in the Materials and Methods, Design section, to further explain this point: “The study protocol was approved by the Institutional Review Board (IRB) of the American University of Beirut. All participants provided verbal informed consent prior to participation in the survey. The IRB waived the requirement for a written informed consent since the survey was anonymous and did not include any identifying data”. 

As for the questionnaire, we submitted to PLOS ONE English and Arabic copies of the Questionnaire used in this study as Supporting Information (S1 and S2). The questionnaire was self-administered by the participants. It took the participants around 15-20 minutes to finish answering all the questions. The open ended questions served the purpose of exploring the reasons that led participants to answer “YES” or “NO” for any particular question. We did not specifically ask participants about the benefits of breastfeeding as this was not the aim of this survey. The questions relating to breastfeeding focused on BFHI steps 3, 4, 5, 6, and 8. Kindly refer to S1 and S2 Appendices for further clarification about the specific questions and their nature.

8. More detail about analysis is also required for both the statistical analysis and the content analysis- how was this completed?

Answer: We amended the section on Data Analysis as follows: “Univariate analysis of participants’ baseline characteristics was conducted by summarizing continuous variables as means and standard deviations, or medians and interquartile ranges as appropriate, and summarizing categorical variables as counts and proportions. Bivariate analysis was conducted to examine the association between knowledge of Baby Friendly hospitals (as a surrogate of knowledge of BFHI practices) and each of the sociodemographic variables, previous breastfeeding experiences, and BFHI practices, using Student’s t test for continuous variables and Pearson Chi-square test for categorical variables. Qualitative data that were generated from open-ended questions were subjected to content analysis and summarized by grouping them under the following themes: what a Baby Friendly hospital is, sources of information/support of breastfeeding, reasons for choosing (or not) to deliver in hospitals that only allow breastfeeding, reasons for choosing (or not) to deliver in hospitals that ban pacifier use, what skin-to-skin contact is, descriptions of a previous skin-to-skin contact experience, reasons for choosing (or not) to repeat/perform skin-to-skin contact in the future, reasons for choosing (or not) to deliver in a hospital that implements skin-to-skin contact, reasons for continuing (or not) to breastfeed a hospitalized infant, reasons for willingness (or not) to pump mother’s own milk to feed her premature newborn, reasons for willingness (or not) to put a premature newborn on the breast to feed her, reasons for willingness (or not) to breastfeed (or pump breast milk) a newborn within one hour from birth, reasons for accepting (or not) to give donor’s milk to the newborn, reasons for willingness (or not) to communicate with a lactation consultant/ breastfeeding support group in future deliveries”.

9. Results: The results section could also be reported in detail. The reader could better understand the findings if it was stated that they are reported with reference to both the quantitative data and the content analysis of the qualitative data to provide a narrative of the key findings. The purpose of Table 2 is clear but it would be helpful, as stated above, if a description of the questionnaire was included in the Methods. This would then give context to the table. The bivariate analysis would benefit from more description. Line 239 refers to knowledge of BFHI practices while the question from the survey asks about knowledge of baby friendly hospitals. This must be consistent as I suggest the two things are not the same. Similarly Table 3 refers to BFHI practices while the question from the survey asks about knowledge of baby friendly hospitals. There is no rationale for why this question from the questionnaire was used for this particular analysis. The aim of the study as stated refers to broader issues than knowing ‘what Baby Friendly hospitals are” so it is not clear why this is the only factor investigated in bivariate analysis. Further explanation is required. 

Answer: Thank you. We incorporated the kind reviewer’s suggestion for easier understanding of the results. We added the following suggested statement at the beginning of page 12: “The remaining quantitative results are reported together with the content analysis of the qualitative data so as to provide a narrative of the key findings”. 

With regards to Table 2 we included with the revised submission English and Arabic copies of the Questionnaire used in this study as Supporting Information (S1 and S2) to make it clearer for the reader.

With regards to the bivariate analysis, as explained in comment #2 above, we chose ‘knowledge of Baby Friendly hospitals’ as a surrogate for ‘knowledge of BFHI practices’ since we know from our daily practice with mothers that the great majority is unfamiliar with Baby Friendly hospitals and the BFHI practices that are implemented in these hospitals.

We corrected the sentence “In bivariate analysis, maternal knowledge of BFHI practices was significantly associated with university education…” to “In bivariate analysis, maternal knowledge of Baby Friendly hospitals was significantly associated with university education..”. Thank you for alerting us to this error.

Table 3: We corrected “Knowledge of BFHI practices” and changed it to “Knowledge of Baby Friendly Hospitals”. Our rationale for using the question about Knowledge of Baby Friendly Hospitals” in the bivariate analysis is because the few women who knew of such hospitals also knew that they implement BFHI practices, and were aware of the BFHI practices. Hence, instead of reporting multiple bivariate analyses to examine the association between each BFHI practice and all of the other variables, we thought that Knowledge of Baby Friendly Hospitals can serve as a surrogate marker for all the BFHI practices. 

10. Discussion: I suggest a further comment on the impact of the obstetrician’s views and the family views. Evidence is provided that shows both to be strongly influential. It would add value if the authors could comment on the possible conflict of differing views by medical staff and family and how a mother may resolve such conflict. Nabulsi et al (2019) describe the importance of combining antenatal breastfeeding education with peer and professional lactation support to improve breastfeeding rates. Perhaps this concept could be expanded in the current paper.

Answer: We added the following paragraph to the Introduction section: “The barriers to the implementation of the BFHI Ten Steps are multifaceted. For example, obstetricians and pediatricians alike tend to easily advise mothers to supplement, or even substitute breastfeeding with artificial milk whenever mothers face any difficulty with breastfeeding. Health care workers in maternity facilities are reluctant to apply the skin-to-skin care immediately after delivery for fear of exposing the baby to cold environments or to accidental fall, as adequate supervision is not always possible. Rooming-in is a challenge because many women prefer to rest after delivery and delegate the care of their newborn to the grand-mother or hospital staff (personal observation), and referral to professional lactation experts or breastfeeding support groups is rarely done. Moreover, women may have several misconceptions regarding breastfeeding initiation and continuation [13,14]”.

We also added the following paragraph to the Discussion section: “A recent study that explored whether breastfeeding initiation and duration would differ by prenatal care provider reported that women who received counseling from a midwife were more likely to exclusively breastfeed as compared to those who received prenatal care from an obstetrician [19]. Hence, there is a need to involve obstetricians in the efforts to promote breastfeeding very early during pregnancy, as part of routine prenatal care. In fact, the American College of Obstetricians and Gynecologists’ current recommendation is for obstetricians to advocate for, and support breastfeeding when offering prenatal care [20]”.

11. Please state why written consent was not required.

Answer: We requested from our IRB a waiver of written informed consent since the survey was anonymous and no identifying data was to be collected from the participants. The IRB approved the waiver of written informed consent. When participants were approached for enrolment in the study, they were informed about the purpose of the study and the type of information that would be collected. They were provided with a written form that summarized the study’s objective and procedure, as well as their rights as research participants and contacts of the research team and the IRB. Please find attached to this rebuttal a copy of the IRB’s approval letter and a copy of the oral consent form. We added the following paragraph to the Materials and Methods, Design section: “The study protocol was approved by the Institutional Review Board (IRB) of the American University of Beirut. All participants provided verbal informed consent prior to participation in the survey. The IRB waived the requirement for a written informed consent since the survey was anonymous and did not include any identifying data”. 

12. It is not clear how the complete data set will be made available to meet the statement: The data will be publicly available for PLOS ONE upon acceptance. 

Answer: We uploaded our anonymized data set as Supporting Information File (S3) on PLOS ONE submission system as part of this revision.

13. Table 1 reports median (SD) for some items. I suggest median (IQR) or mean (SD).

Answer: Thank you. (SD) was replaced with (IQR) for medians.

14. Line 23 Introduction: I suggest changing the term designated as Baby Friendly to accredited as Baby Friendly – unless there is a reason for using the term designated.

Answer: The term ‘designated’ was replaced by ‘accredited’ in all the manuscript.

15. Line 25 Aim: I suggest a rearrangement of the aim to improve understanding. The aim stated in the manuscript lines 116-117 is structured more appropriately. Please consider using this to replace the current aim in the abstract. 

Answer: The Aim in the Abstract was changed as suggested. It now reads as follows: “To assess the knowledge of Lebanese pregnant women of BFHI steps, and to explore their attitudes towards Baby Friendly Hospitals, and acceptance of Skin-to-Skin Contact and Kangaroo Care practices”.

16. Abstract Results: minor grammatical errors.

Answer: The grammatical errors in the Abstract Results were corrected.

17. Line 40-41: the phrase “and their numerous advantages” does not make it clear what ‘their” is referring to and thus could be misleading. Suggest: There is a need for national awareness campaigns that address both the numerous advantages of the BFHI practices and Lebanese women’s knowledge gaps about these practices.”

Answer: We rephrased the sentence as suggested.

18. Line 42: suggest adding and thus “through increased breastfeeding” positively impact 

Answer: We rephrased the sentence as suggested.

19. Please consider including the ideas expressed in lines 118-121 as part of the conclusion of the abstract.

Answer: We revised the conclusion as follows: “There is a need for national awareness campaigns that address both the numerous advantages of the BFHI practices and Lebanese women’s knowledge gaps about these practices. Such knowledge will help scale up the implementation of BFHI practices in hospitals in Lebanon, thus increasing breastfeeding rates and positively impacting the health of infants and mothers”. 

20. Line 129 states clinics while line 130 states licensed obstetricians. Should clinics be the consistently used term here for consistency? As part of this description please state the process for random identification of the clinics and also the total number of clinics from which the 50 clinics were chosen.

Answer: We used ‘clinics’ throughout this section for consistency. We revised this paragraph as follows: “Obstetric Clinics were chosen randomly from the list of the Lebanese Society of Obstetrics and Gynecology. The list includes the addresses and contacts of the 258 obstetric clinics from all six governorates (Mouhafaza) of Lebanon: Greater Beirut (Lebanon’s capital), North Lebanon, South Lebanon, Mount Lebanon, Nabatiyeh, and Beqaa. A computer-generated random sampling of 20% of these clinics was conducted by an independent biostatistician. Thus 25 clinics from Greater Beirut, and 5 clinics from each of the remaining 5 governorates were included in this study. This distribution was chosen so as to parallel the distribution of clinics in the different governorates of Lebanon”. 

21. Line 138: please state the criteria for consecutive recruitment eg. When did this recruitment start and was it the same time frame for each clinic?

Answer: We revised the Methods, Sample as follows: “Recruitment of participants started in April 2016 and ended in March 2018. A trained research assistant contacted the randomly chosen clinics in each Mouhafaza to arrange for appointments to visit clinics for recruitment of participants. From each clinic, ten healthy pregnant women who were 18 years of age and older were consecutively recruited as they presented to the clinic. Recruitment of participants was happening in the same time frame for clinics located in the same Mouhafaza. The total sample size for this survey was 500 participants”. 

22. Line 141: please clarify why verbal and not written consent was obtained. Also clarify if the questionnaire was written, and if self-administered, did the participant take the questionnaire away or answer in the clinic? Was there any involvement of the staff? How long was the questionnaire expected to take and how many questions and what type of responses were included. Later in the section open ended questions are described. Generally more detailed information is required in this section. “Participants were asked” may be misleading, could you rephrase to “The questionnaire asked participants about..”

Answer: We requested from our IRB a waiver of written informed consent since the survey was anonymous and no identifying data was to be collected from the participants. The IRB approved the waiver of written informed consent. When participants were approached for enrolment in the study, they were informed about the purpose of the study and the type of information that would be collected. They were provided with a written form that summarized the study’s objective and procedure, as well as their rights as research participants and contacts of the research team and the IRB. Please find attached to this rebuttal a copy of the IRB’s approval letter and a copy of the oral consent form. We added the following paragraph to the Materials and Methods, Design section: “The study protocol was approved by the Institutional Review Board (IRB) of the American University of Beirut. All participants provided verbal informed consent prior to participation in the survey. The IRB waived the requirement for a written informed consent since the survey was anonymous and did not include any identifying data”. 

The questionnaire was written and the participants answered it in the clinic, with no involvement of the staff. It took the participants around 15-20 minutes to finish answering all the questions. As for the details of the questions, we submitted to PLOS ONE English and Arabic copies of the Questionnaire used in this study as Supporting Information (S1 and S2). The open ended questions served the purpose of exploring the reasons that led participants to answer “YES” or “NO” for any particular question. Kindly refer to Supporting Information file S1 Appendix for the closed and open-ended questions.

We revised the sentence ‘Participants were asked..” to ‘The questionnaire asked participants about..’ as suggested by the kind reviewer.

23. Line 150: were participants briefed after they completed the first part of the questionnaire? Please make explicit and include a statement about who briefed the participants eg. Obstetricians etc..

Answer: The trained research assistant briefed the participants about the benefits of skin-to-skin contact and early breastfeeding after they completed the first part of the questionnaire (after answering question 14 in S1). This information is now added to the revised manuscript as follows: “Consequently, the trained research assistant briefed the participants about early breastfeeding and skin-to-skin contact benefiting the sick preterm infant health and shortening hospital stay. The participants then continued answering the remaining questions in the questionnaire that asked whether they would do skin-to-skin contact..”. 

24. Line 152: again please clarify if these questions were verbally asked or if the statement refers to the written questions in the survey.

Answer: All questions were written and not verbal. This sentence is now revised as follows: “The participants then continued answering the remaining questions in the questionnaire that asked whether they would..”

25. Line 157: I note medians are also included in Table 1 but not stated in this section.

Answer: Thank you. This is now revised as follows: “Univariate analysis of participants’ baseline characteristics was conducted by summarizing continuous variables as means and standard deviations, or medians and interquartile ranges as appropriate, and summarizing categorical variables as counts and proportions”.

26. Line 255: Please clarify if it is only mothers with poor knowledge who would birth their baby in a Baby Friendly hospital next time. I suspect this is about the phrasing of the sentence rather than the data as the next sentence describes this. Can you change the order of the sentences to make this clearer?

Answer: This paragraph is revised as suggested to make it clearer to the reader: “Participants with knowledge of BFHI and a previous experience of skin-to-skin contact, or rooming-in, or those who were well-instructed about breastfeeding were very willing to repeat the same experience in the future. Interestingly, most mothers with poor knowledge of BFHI practices stated that they would deliver their future infants in Baby Friendly hospitals after they were provided with the information about the benefits of these practices”.

27. Line 313: Comment as for the conclusion in the abstract. The authors provide evidence that family and peers may be influential in breastfeeding practices and thus I suggest this is also addressed in terms of increasing their knowledge.

Answer: The conclusion is revised as follows: “In conclusion, this study highlights the need for national awareness campaigns that address both the numerous advantages of the BFHI practices and Lebanese women’s knowledge gaps about these practices. Such knowledge will facilitate the implementation of BFHI in Lebanon as women may become more engaged in these practices, and will seek to deliver in Baby Friendly hospitals thus encouraging maternity facilities to adopt the BFHI practices. Moreover, there is a need to spread the knowledge of BFHI practices to the public at large since family and peers may be influential in breastfeeding practices, as well as to obstetricians, pediatricians and family physicians that are the primary source of medical information for patients”.

Response to Reviewer #4

Thank you for taking the time to review our manuscript. Please find below our point by point response. 

1. As the research was done in a Middle East country, I think some articles related to BFHI in other countries should be added. In lines 77-78 just mentioned about Canada. How about Middle East countries?

Answer: In the Introduction, we added information about BFHI implementation in different areas of the world, including Middle East countries from the WHO 2017 report: “Recently, the WHO reported an analysis of the current BFHI status in 168 countries around the world after 25 years of BFHI initiation [5]. As of 2016, the estimated overall BFHI coverage is 10%, with the highest rate (36%) being in the European region, and the lowest (< 5%) in Africa and Southeast Asia. The Eastern Mediterranean region (EMR) has an intermediate coverage of 17% that is largely driven by a few countries with large populations. Only one in five countries had ever accredited more than half of their facilities as Baby Friendly. For Lebanon, this rate declined from 18% to 10% in the past 5 years [5]. There is a great variation in BFHI implementation in the EMR countries based on the countries’ categorization with regards to nutrition stages. Countries in the advanced nutrition stage like the Gulf Cooperation Council have 80% of their hospitals accredited as Baby Friendly, whereas countries in early nutrition transition like Lebanon have only 6.85% of their hospitals as Baby Friendly [6]”.

2. BFHI was launched by WHO and UNICEF that have ten steps, and step 4 of the Ten steps recommends that “facilitates immediate skin to skin contact”. When this paper aimed to assess the knowledge of women about BFHI steps and skin to skin contact is the 4th step, explore about skin to skin contact separately can be meaningless.

Answer: Our questionnaire targeted all the BFHI steps. Please refer to the English questionnaire that is now submitted to PLOS ONE as Supporting Information S1 Appendix. We specifically targeted skin-to-skin contact or kangaroo care practices in this paper because most mothers in Lebanon have never heard of skin-to-skin-contact or kangaroo mother care, and have never practiced rooming in. Moreover, when we tried to convince some hospital administrators to implement these practices we were told that Lebanese women would not ‘like’ such practices as they may find them threatening to the well-being of their newborns. This was one of the major reasons why we conducted this survey as evidence to the contrary argument: that women would welcome such practices if they knew of their advantages. In the Introduction, we added the following sentence as justification for focusing on step 4: “We have observed that most Lebanese women are unfamiliar with the term Baby Friendly Hospital, and are unaware of its practices, especially the Skin-to-Skin Contact or Kangaroo Care practices.” 

3. What is including criteria? In tables we are not able to see how many women were primipara.

Answer: The inclusion criteria are stated in Methods, Sample: “.. healthy pregnant women who were 18 years of age and older..” visiting the clinics that were randomly chosen from all Lebanon. The number of primiparous women was 231. In Table 1 footnote, we indicated the number of multiaprous participants (N=286) from which one can calculate the number of primiparous participants.

We hope we have addressed all the comments of the kind reviewers in a clear manner, and revised the manuscript accordingly.

Thank you again for the thorough review of our paper.

Best wishes.

Yours sincerely, 

Mona Nabulsi, MD, MSc

Professor of Clinical Pediatrics

Department of Pediatrics and Adolescent Medicine

Faculty of Medicine

American University of Beirut

Beirut-Lebanon

P.O.Box: 113-6044/C8

Fax: 961-1-370781

 961-1-744464

E-mail: mn04@aub.edu.lb

P.O.BOX 11-0236 RIAD EL-SOLH – BEIRUT 1107 2020 – LEBANON – TEL (961 1 374374 – EXT:5500) FAX (961 1 370781 or 744464)

E-MAIL:peds@aub.edu.lb-NEW YORK OFFICE: 850 THIRD AVE.18TH FLOOR–NEW YORK, NY 10022-6297 U.S.A.-TEL. (1 212 583 7677)

---

## [Decision Letter · Decision Letter 1]

20 Aug 2020

PONE-D-20-05601R1

Knowledge and attitudes of Lebanese women towards Baby Friendly Hospital Initiative practices

PLOS ONE

Dear Dr. Nabulsi,

Thank you for submitting your manuscript to PLOS ONE. We invite you to submit a revised version of the manuscript that addresses the points raised during the review process.

We look forward to receiving your revised manuscript.

Kind regards,

Thach Duc Tran, M.Sc., Ph.D.

Academic Editor

PLOS ONE

Reviewers' comments:

Reviewer's Responses to Questions

**Comments to the Author**

1. If the authors have adequately addressed your comments raised in a previous round of review and you feel that this manuscript is now acceptable for publication, you may indicate that here to bypass the “Comments to the Author” section, enter your conflict of interest statement in the “Confidential to Editor” section, and submit your "Accept" recommendation.

Reviewer #1: All comments have been addressed

Reviewer #2: All comments have been addressed

Reviewer #3: All comments have been addressed

2. Is the manuscript technically sound, and do the data support the conclusions?

Reviewer #1: (No Response)

Reviewer #2: Yes

Reviewer #3: Yes

3. Has the statistical analysis been performed appropriately and rigorously? 

Reviewer #1: (No Response)

Reviewer #2: Yes

Reviewer #3: I Don't Know

4. Have the authors made all data underlying the findings in their manuscript fully available?

Reviewer #1: (No Response)

Reviewer #2: Yes

Reviewer #3: Yes

5. Is the manuscript presented in an intelligible fashion and written in standard English?

Reviewer #1: (No Response)

Reviewer #2: Yes

Reviewer #3: Yes

6. Review Comments to the Author

Reviewer #1: (No Response)

Reviewer #2: The Author replied and revised all of my comments and in fact the Author satisfied me. Thank you for this.

Reviewer #3: Thank you for the detailed and comprehensive responses to the points raised in the review. This has significantly improved the manuscript and it now represents your research in a clear and articulate manner. One minor point is that in line 186 the word "Consequently" is probably not the appropriate term. I suggest something like 'Once participants had responded to questions xxx the trained research assistant..." Otherwise - very well done.

7. PLOS authors have the option to publish the peer review history of their article (what does this mean?). If published, this will include your full peer review and any attached files.

Reviewer #1: No

Reviewer #2: No

Reviewer #3: No

---

## [Author Response · Author response to Decision Letter 1]

20 Aug 2020

Response to Reviewer #3

We greatly appreciate your comments and valuable feedback. Please find below our response. 

1. Line 186: “Consequently” is probably not the appropriate term. I suggest something like “Once participants had responded to questions xxx the trained research assistant..”.

Answer: Thank you for the suggested edit. We revised the sentence as suggested: “Once participants had responded to the aforementioned questions, the trained research assistant briefed them about….”.

Thank you for the thorough review of our paper.

Best wishes.

---

## [Editor Report · Decision Letter 2]

24 Aug 2020

Knowledge and attitudes of Lebanese women towards Baby Friendly Hospital Initiative practices

PONE-D-20-05601R2

Dear Dr. Nabulsi,

We’re pleased to inform you that your manuscript has been judged scientifically suitable for publication and will be formally accepted for publication once it meets all outstanding technical requirements.

Kind regards,

Thach Duc Tran, M.Sc., Ph.D.

Academic Editor

PLOS ONE
---

## [Editor Report · Acceptance letter]

31 Aug 2020

PONE-D-20-05601R2 

Knowledge and attitudes of Lebanese women towards Baby Friendly Hospital Initiative practices 

Dear Dr. Nabulsi:

I'm pleased to inform you that your manuscript has been deemed suitable for publication in PLOS ONE. Congratulations! Your manuscript is now with our production department. 

Kind regards, 

on behalf of

Dr. Thach Duc Tran 

Academic Editor

PLOS ONE